# A Common Target of Nitrite and Nitric Oxide for Respiration Inhibition in Bacteria

**DOI:** 10.3390/ijms232213841

**Published:** 2022-11-10

**Authors:** Wei Wang, Jiahao Wang, Xue Feng, Haichun Gao

**Affiliations:** Institute of Microbiology, College of Life Sciences, Zhejiang University, Hangzhou 310058, China

**Keywords:** cytochrome *c*, cytochrome *c* biosynthesis, nitrite, nitric oxide, respiration

## Abstract

Nitrite and nitric oxide (NO) are well-known bacteriostatic agents with similar biochemical properties. However, many studies have demonstrated that inhibition of bacterial growth by nitrite is independent of NO. Here, with *Shewanella oneidensis* as the research model because of its unusually high cytochrome (cyt) *c* content, we identify a common mechanism by which nitrite and NO compromise cyt *c* biosynthesis in bacteria, and thereby inhibit respiration. This is achieved by eliminating the inference of the cyclic adenosine monophosphate-catabolite repression protein (cAMP-Crp), a primary regulatory system that controls the cyt *c* content and whose activity is subjected to the repression of nitrite. Both nitrite and NO impair the CcmE of multiple bacteria, an essential heme chaperone of the System I cyt *c* biosynthesis apparatus. Given that bacterial targets of nitrite and NO differ enormously and vary even in the same genus, these observations underscore the importance of cyt *c* biosynthesis for the antimicrobial actions of nitrite and NO.

## 1. Introduction

Cytochromes *c* (cyts *c*), ubiquitous heme-containing proteins present in all domains of life, are essential for respiration and photosynthesis [1]. This type of protein is characterized by the covalent thioether bonds between the two cysteine residues within a heme-binding motif (HBM) (typically CXXCH, where X represents any amino acid residue) of apocytochrome (apocyt) and the two vinyl groups of the heme [2]. Multiple enzymatic complexes, called cyt *c* biosynthesis systems, are known to catalyze the formation of the bond [3]. Among them, System I, also called the cyt *c* maturation (CCM) system, found in diverse Gram-negative bacteria and archaea, as well as in plant and protozoan mitochondria, is no doubt the most complex and has been extensively studied in certain Gram-negative bacteria [2,3,4,5,6].

System I is composed of up to nine protein components, which form two functional modules, CcmABCDE for heme transport and delivery and CcmFGH(I) for disulfide reduction and heme-apocyt *c* ligation [2,7]. In Gram-negative bacteria, cyt *c* biosynthesis occurs entirely in the periplasm, requiring both the apocyt *c* and heme *b* molecules, which are translated and synthesized, respectively, in the cytoplasm and, subsequently, transported across the inner membrane (IM). While the translocation of apocyt *c* is a CCM-independent process carried out by the general secretion system, the CcmABCD complex mediates the transport of heme *b* into the periplasm. As a periplasmic heme chaperone, CcmE receives the heme from CcmABCD and subsequently passes it to the CcmFGH(I) module [8,9,10,11]. During heme transport, heme is attached to CcmE through a single covalent bond between a histidine (H130 in CcmE of *Escherichia coli*) and a vinyl group of heme [12,13]. Both the apocyt *c* and heme *b* molecules are eventually loaded onto the CcmFGH(I) module for ligating heme to apocyt *c* [7,14,15].

Nitrite and nitric oxide (NO), ubiquitous and highly toxic nitrogen oxides, can damage cells by directly reacting with the redox-active proteins, for instance with heme to form an intermediate species with a ferrous-nitrogen dioxide character [16,17,18,19]. Nitrite has been traditionally used as a food preservative to inhibit the growth of undesirable bacteria for centuries, but its bacteriostatic effects are commonly attributed to NO [19,20]. However, recent studies have suggested that cellular targets of nitrite and NO are largely distinct, varying substantially even in the bacterial isolates in the same species [21,22,23,24]. Similarly, detoxification of nitrite and NO is carried out by different systems in bacteria [25]. While a large group of diverse enzymes catalyzes the transformation of nitrite to less-toxic nitrogen species, such as nitrate, ammonium, and dinitrogen gas, NO is primarily degraded by oxygen-dependent flavohemoglobin [26]. Moreover, proteins responsible for sensing nitrite and NO and mediating cellular responses to the resulting nitrosative stresses are also different, implying that bacterial cells treat these two nitrogen oxides distinctly in terms of physiology [25,26]. Thus, whether there is a common mechanism by which nitrite and NO influence bacterial physiology remains unknown.

*Shewanella oneidensis*, a Gram-negative γ-proteobacterium renowned for respiratory versatility, has served as a research model for extracellular electron transfer, a process that is largely based on a large repertoire of cyts *c* (up to 42) [5,27,28,29]. This unusually high cyt *c* content confers *S. oneidensis* cells (colony and cell pellet) a red-brown color, and given that the color intensity correlates well with the abundance of cyt *c*, it can be exploited to conveniently identify factors affecting the cyt *c* content [6,23,24,30,31]. More importantly, with this bacterium as a research model, we have demonstrated that cyts *c* constitute a defense frontline protecting metabolic enzymes from NO damage, whereas the primary targets of nitrite for oxygen respiration are heme-copper oxidase (HCO), including cyt *aa*_3_ as in *Bacillus subtilis*, cyt *bo*_3_ as in *E. coli*, and cyt *cbb*_3_ as in *S. oneidensis* and *Pseudomonas aeruginosa* [23,24,25,32,33].

During our investigation, we noted that nitrite down-regulates the overall cyt *c* content, especially when fumarate is used as the electron acceptor (EA) to support growth [34]. This effect involves the cyclic adenosine monophosphate-catabolite repression protein (cAMP-Crp) transcriptional regulatory system, which mediates the transcription of a large number of genes encoding proteins involved in respiration, including many cyts *c* [32,34,35,36]. In this study, we first substantiate that the reduction in cAMP levels caused by nitrite or NO is not fully responsible for the decrease in the cyt *c* content in *S. oneidensis*. We then show that, although heme is implicated, its biosynthesis and cellular levels are not significantly affected by both of the nitrogen oxides. Further investigations reveal that nitrite and NO interfere with heme transport by compromising CcmE, leading to impaired cyt *c* biosynthesis.

## 2. Results

### 2.1. Nitrite and NO Down-Regulate cyt c Content of S. oneidensis

We found by chance that nitrite significantly reduces overall cyt *c* abundance in *S. oneidensis* wild-type (WT) cells grown on various EAs, fumarate in particular, as the cell pellets became evidently paler when nitrite was present [34] (Figure 1A,B). Given that nitrite shares similar biochemical properties with NO [19,26,37], we tested whether NO is also able to down-regulate the cyt *c* content in *S. oneidensis*. Indeed, a significant reduction in cyt *c* abundance was observed in cells grown in the presence of NO-releasing agent, NOC-18 (DETA-NONOate, t_1/2_ ≈ 20 h), at proper levels (Figure 1B and Appendix A). When NO scavenger carboxy-PTIO at 0.1 mM was added, which was able to quench the NO signal from NOC-18 to levels below the measurable limit, the cyt *c* content was not significantly altered (Figure 1B), validating the association of NO with the cyt *c* content. The application of carboxy-PTIO failed to prevent the cyt *c* content reduction in cells grown with nitrite (Figure 1B), eliminating the possibility that the observed effect of nitrite on the cyt *c* content is via NO. These data suggest that both nitrite and NO, independently of each other, are able to down-regulate the cyt *c* content in *S. oneidensis*.

In order to unravel the underlying mechanism for the effect, we compared cyt *c* profiles in the extracts of *S. oneidensis* WT cells grown on fumarate without or with nitrite at varying concentrations by heme staining. Nitrite at concentrations no less than 2 mM showed a negative effect on the abundance of all cyts *c*, including fumarate reductase FccA and nitrite reductase NrfA, verified by gene-deletion mutants and mass spectrometry (MS) analysis previously [6,38,39] (Figure 1C). Similar results were obtained from NO, whose effects on the cyt *c* content became evident when it was generated from NOC-18 at 50 μM or higher (Figure 1D). It should be noted that the effects of nitrite and NO on the cyt content can only be assessed non-quantitatively as the minimal inhibitory concentrations of these two nitrogen oxides on growth differ substantially [23,24]. Despite this, the reduced overall amounts of cyts *c* in the presence of either nitrite or NO at high concentrations were consistent with the data of heme *c* quantification (Figure 1B). These results conclude that both nitrite and NO impact the cyt *c* content, likely through a similar mechanism, and we therefore use nitrite/NO to simplify the description hereafter.

### 2.2. Nitrite/NO Compromise the cyt c Content in Growing Cells Only

It is fully established that hemoproteins are the primary targets of nitrite/NO by in vitro analyses [1,17,23,26,37]. We therefore moved on to test whether the reduction in the cyt *c* content caused by nitrite/NO is a result of cyt *c* damage and/or destruction. To this end, WT cells grown to the early stationary phase were collected and disrupted by sonication, and the cell debris was treated with nitrite/NO at different concentrations for 5 h. Even with 10 mM nitrite or NOC-18, the cyt *c* content of the samples was not found to be significantly altered before and after the treatments (Figure 2A and Appendix A), suggesting that nitrite/NO are unlikely able to release heme molecules from cyts *c*. To confirm this, bovine heart cyt *c* acquired commercially was incubated for 5 h with nitrite and NOC-18 at concentrations up to 120 mM and 10 mM, respectively. It was immediately evident that NOC-18 at 10 mM, but not nitrite at 120 mM, showed a visible influence on the color of the bovine heart cyt *c* solution, which became more pinkish (Appendix A). Treated cyt *c* was then examined by SDS-PAGE/heme staining and spectroscopic measurements. As shown in Figure 2B, the migration of this protein was found unaffected by nitrite at all concentrations under test. However, although the treatment with NOC-18 at concentrations 5 mM or less was not able to elicit a detectable difference in bovine heart cyt *c*, a portion of the protein appeared to be modified with 10 mM NOC-18, probably due to the formation of the nitrosyl complex [17]. This was validated by the UV–visible absorption analysis. The spectrum of cyt *c* can be characterized by absorption of γ- (405 nm) and α- (550 nm) Soret bands. Solutions of this cyt *c* in ferric form exhibited a Soret absorption at 405 nm with an unresolved α band at 550 nm (Figure 2C). The addition of 10 mM NOC-18 generated a Soret absorption at 415 nm and evidently a band at 550 nm, which is typical for ferrous cyt *c*. The shift in Soret absorption was from 405 to 415 nm, which is the signature of the Fe(III)-nitrite complex [40,41]. In contrast, the cyt *c* treated by up to 20 mM nitrite was not different in the UV–visible absorption spectra from the untreated control (2D).

Despite the observed modification of cyt *c* by NO of high concentrations, it is unlikely the reason for the substantial reduction in the cyt *c* content in living cells. To provide evidence, we assessed whether the effect of nitrite/NO can only be observed from viable cells. Cells prepared as above were subjected to either gentamycin (Gent) sulfate or ether treatment. Although both Gent- and ether-treated cells are non-viable, but intact, the latter differ from the former in that they can still carry out many biological processes, such as DNA synthesis and peptidoglycan synthesis, provided that the required substrates are supplemented [42,43]. Despite this difference, after incubation with 10 mM nitrite/NOC-18 for 5 h, the cyt *c* contents in samples before and after each of the treatments were not significantly different (Figure 2A and Appendix A), suggesting that the impact of nitrite/NO is not effective with respect to these non-viable cells. Based on all of these data, we concluded that, despite their inhibitory effects on cyt *c* activity, nitrite/NO compromise the overall cyt *c* content only in viable cells because they are unable to break the covalent bond between heme molecules and the peptide at the physiologically relevant concentrations.

### 2.3. cAMP-CRP Is Not Exclusively Responsible for the Nitrite-/NO-Mediated Reduction in the cyt c Content

As nitrite/NO reduce the cyt *c* content not by directly disrupting cyts *c* per se, we reasoned that these molecules may down-regulate the expression of a significant share, if not all, of the genes for cyts *c*. In *S. oneidensis*, it is well established that the compromised activity of cAMP-Crp, which is the master regulatory system for respiration and directly controls transcription of a large portion of cyt *c* genes, results in a reduced cyt *c* content [32,33,34,35,44,45,46]. More importantly, nitrite has been implicated in down-regulating intracellular concentrations of cAMP, leading to lowered activity of cAMP-Crp [34]. However, whether the reduction in the cAMP levels exclusively accounts for the reduced content of cyts *c* caused by nitrite is unknown.

We reasoned that this can be assessed with Crp mutants that no longer require cAMP for activity. In *E. coli*, a Crp mutant, *Ec*Crp^T128L-S129I^, showed a cAMP-independent DNA binding affinity comparable with that of cAMP-bound wild-type *Ec*Crp [47]. In accord with the high sequence similarity (BLASTp E-value, 5e-139) between *E. coli* and *S. oneidensis* Crp proteins, *Ec*Crp has been shown to be functional in *S. oneidensis* [35]. Despite this, the specific regulatory activity of *Ec*Crp in *S. oneidensis* was determined with well-characterized CRP-dependent promoters, P*_CC_*_(−*41.5*)_ and P*_cyd_*. While P*_CC_*_(−*41.5*)_ is a semisynthetic derivative of the *melR* promoter of *E. coli*, P*_cyd_* directs transcription of the *cyd* operon encoding cyt *bd* oxidase in *S. oneidensis* [32,48]. DNA fragments for all Crp variants were placed under IPTG-inducible promoter P*_tac_* within expression vector pHGE-Ptac used repeatedly in *S. oneidensis* [49]. As expected, *Ec*Crp and *Ec*Crp^T128L-S129I^, the same as *So*Crp, could activate the expression of both promoters in the Δ*crp* strain in the presence of 0.1 mM IPTG (Figure 3A). Such an activating effect was still observed with *Ec*Crp^T128L-S129I^ in the absence of cAMP (Δ*crp*Δ*cya,* lacking all three genes for adenylate cyclases (ACs)), and in contrast, neither P*_CC(_*_−*41.5)*_ nor P*_cyd_* were responsive to the induced expression of *Ec*Crp and *So*Crp with up to 0.5 mM IPTG (Appendix A). These data validate that *Ec*Crp^T128L-S129I^ could function in a cAMP-independent manner as a transcriptional activator in *S. oneidensis*.

When grown on oxygen, both *crp* and *cya* mutants contained significantly decreased cyt *c* contents, approximately by 40% relative to the wild-type level (Figure 3B). The defect in the cyt *c* content of the Δ*crp* strain can be fully corrected by the expression of *So*Crp, *Ec*Crp, or *Ec*Crp^T128L-S129I^ with 0.1 mM IPTG. However, *Ec*Crp^T128L-S129I^, but not *Ec*Crp or *So*Crp, produced with IPTG 0.1 mM and above fully restored the cyt *c* content of Δ*crp*Δ*cya* to the wild-type level (Figure 3B). Despite this, nitrite/NO were still able to significantly reduce the cyt *c* content of Δ*crp*Δ*cya* expressing *Ec*Crp^T128L-S129I^ (Figure 3B). Based on this, we concluded that the nitrite-/NO-mediated reduction in the cyt *c* content in *S. oneidensis* could not be attributed to decreased cAMP levels exclusively.

### 2.4. Increased Heme Levels Alleviate the Effect of Nitrite/NO on the cyt c Content

cyts *c* differ from other hemoproteins in that their heme *b* molecules are covalently attached to HBMs [2]. Given that heme is a substrate of cyt *c*, we hypothesized that the ligand may be a factor implicated in the reduced cyt *c* content caused by nitrite/NO. *S. oneidensis* possessing the common pathway for heme synthesis (Figure 4A), which entails nine reactions that convert glutamyl-tRNA to protoporphyrin IX [46,50]. HemA (glutamyl-tRNA reductase) catalyzes the first dedicated, rate-limiting step in heme synthesis [51] (Figure 4A). To test whether nitrite/NO affect the synthesis of heme, we assessed the impacts of nitrite/NO on the expression of the *hemA* gene, as well as two other *hem* genes and found that the impact was insignificant (Appendix A). In line with this, we found that the nitrite/NO supplement did not significantly alter the heme levels in the cells (Appendix A). Nevertheless, we continued to examine whether increased heme production could counteract the effect of nitrite/NO on the cyt *c* content. As 5-ALA is used exclusively for heme production and its synthesis is a critical focal point for the regulation of heme biosynthesis [51], we first determined if exogenous 5-ALA influences the cyt *c* contents of *S. oneidensis* cells. In line with the previous observations [52], the addition of 50 µM 5-ALA induced a noticeable increase in cyt *c* abundance, and a 35% increase was observed with 200 µM ALA (Figure 4B). A similar effect of 5-ALA addition was observed in the presence of nitrite/NO despite the overall lowered cyt *c* levels.

The improving effect of elevated heme levels on the cyt *c* content was further assessed with manipulated *hemA* expression. In cells grown under normal conditions, the activity of the *hemA* promoter was not higher than that of P*_tac_* with 0.05 mM IPTG [23,46]. As shown in Figure 4B, HemA overproduction with 0.2 mM IPTG was able to improve cyt *c* content by approximately 30%, but no further increase was observed with higher concentrations. Importantly, the effect of HemA in excess was also evident when 5.0 mM nitrite or 0.1 mM NOC-18 was present (Figure 4B and Appendix A). These results suggest that heme may have a role in nitrite-mediated reduction in cyt *c* content.

### 2.5. Identification of CcmE as a Likely Target of Nitrite/NO on cyt c Production

To elucidate the involvement of heme in the effect of nitrite/NO on cyt *c* production, we turned to the CCM system. *S. oneidensis* possesses a highly conserved CCM system for cyt *c* biosynthesis, but its components are arranged into functional modules encoded by three operons, *ccmABCDE* (heme delivery), *ccmI*, and *CcmFGH* (heme-apocyt *c* ligation), in the *ccm* locus, in contrast to a single operon for the *ccm* genes present in most other γ-proteobacteria [4,6] (Appendix A). After being synthesized in the cytoplasm, the apocyts *c* and heme *b* molecules are exported to the periplasmic compartment via the classical Sec protein secretion apparatus and by the CcmABCDE module, respectively, where the ligation between heme and HBM occurs [11,14] (Figure 5A). To test which functional module is implicated in nitrite-mediated reduction in cyt *c* content, we overexpressed each of them in WT cells. As shown in Figure 5B, in the presence of 5.0 mM nitrite, the lowered cyt *c* content could not be rescued by overproduction of either CcmFGH or CcmI. In contrast, overproduction of CcmABCDE was able to modestly improve cyt *c* production when nitrite was present, suggesting that heme delivery may be subject to nitrite inhibition. Further investigations showed that the overproduction of CcmE, but not CcmABCD was able to improve the production of cyt *c* when nitrite was present, implying that nitrite may affect the physiological function of CcmE (Figure 5B).

For confirmation, we constructed a *ccmE* mutant and monitored how the cyt *c* content changed in response to CcmE produced at varying levels. The results revealed that the Δ*ccmE* strain was indistinguishable from the Δ*ccmF* strain and the WT when a copy of *ccmE* was expressed in trans with IPTG at 0.2 mM and above (Appendix A). Clearly, CcmE is essential for cyt *c* maturation in *S. oneidensis*, but it would not enhance the overall cyt *c* content when overproduced alone (Figure 5B). However, in the presence of nitrite, the effect of overproduced CcmE became evident: it antagonized the inhibition of nitrite (Figure 5C). Similar results were observed with NO (Appendix A). Moreover, we tested whether nitrite would impact the activity of CcmE proteins in general. When CcmE proteins of *E. coli* and *P. aeruginosa* were overexpressed in the Δ*ccmE* strain, both were found to be able to counteract the effect of nitrite on the cyt *c* biosynthesis to some extent, albeit not so effective as the *S. oneidensis* counterpart (Figure 5C and Appendix A). Altogether, these data suggest that the reduced production of the cyt *c* content in the presence of nitrite/NO is, at least in part, due to the compromised CcmE activity.

### 2.6. Nitrite/NO Impair CcmE Activity through a Complex Mechanism

Among all System I components, CcmE is characterized by a unique feature in which it binds heme by way of a single covalent bond to a histidine [8]. As this covalent linkage also differs from the double covalent bonds observed in cyts *c*, we hypothesized that the linkage may be sensitive to nitrite/NO. To test this, we overexpressed His_6_-tagged CcmE recombinant proteins in the Δ*ccmE* background and examined their stability before and after nitrite/NO treatment. The recombinant protein, which was functional because it restored the cyt *c* biosynthesis in Δ*ccmE*, can be detected by heme staining and Western blotting (Appendix A). Compared to control samples, cells grown with nitrite at 10 mM or NOC-18 at 0.1 mM showed significant reduction in the amount of the recombinant CcmE detectable by heme staining (Figure 6A). However, Western blotting revealed that the treatment did not critically affect the amounts of protein detected. These data imply that the treatment does not reduce the quantity of CcmE, but more likely modifies the proteins, making them undetectable by the classical heme staining approach.

We then expressed and purified the recombinant CcmE protein for UV–visible spectroscopy characterization before and after the nitrite/NO treatment. Compared to solutions of ferric CcmE, which had a Soret absorption at 405 nm, the samples treated either by 10 mM nitrite or 5 mM NOC-18 exhibited an absorption shift from 405 to 415 nm, along with absorption at 550 nm, indicating that both nitrogen oxides are able to modify CcmE (Figure 6C,D). Notably, the modification of CcmE by NO occurred at a concentration four-times lower than that required for cyt *c* modification (Figure 2D). These data suggest that CcmE is substantially more susceptible to modification by nitrite and NO than cyt *c*.

## 3. Discussion

Because of its bacteriostatic effect, nitrite has been used as a preservative in meat products for centuries. About 40 years ago, it was proposed that the bacteriostatic effect of nitrite is attributed to NO formation [20,53]. This appears reasonable, as in vitro studies show that nitrite and NO display similar, albeit not identical, biochemical properties, and therefore, the proteins susceptible to them are similar, mainly those containing redox-active centers such as heme, iron-sulfur clusters, mono-/di-nuclear iron, thiol, and so on [17,19,26,54]. In accord with this, most of the cellular targets identified in vivo to date are metabolic and respiratory enzymes depending on their redox-active centers for catalysis and/or oxi-reduction [55]. However, more recent investigations have revealed that key cellular targets of nitrite and NO in the bacterial species studied are largely different [22,23,24,25,55]. While the NO targets are primarily metabolic enzymes in the cytoplasm, nitrite specifically inhibits all types of HCOs, which are membrane-bound complexes exposed to the extracellular space [25]. This may be readily explained by the fact that NO diffuses into the cytoplasm freely, whereas nitrite is a charged and, therefore, membrane-impermeable molecule and has to rely on specific transporters to enter the cytoplasm in many bacteria [56,57]. It is therefore conceivable that the concentrations of nitrite on both sides of the membrane would make a difference.

Despite this, we continued in our endeavor to search for common targets of nitrite and NO in bacteria. In this study, by taking advantage of the high abundance of cyts *c* in *S. oneidensis* cells, we identified CcmE to be the one susceptible to both nitrite and NO. Although *S. oneidensis* is renowned for respiratory versatility, it has a limited capacity in dealing with nitrite [58]. Nitrite can only be transformed to ammonia without releasing any intermediate, and critically, this process occurs only in the absence of oxygen [59]. Similarly, the NO physiology of *S. oneidensis* is rather simple. This bacterium lacks not only an enzymatic source for NO generation, either bacterial nitric oxide synthase or denitrifying system, but also an efficient NO scavenger, such as flavohemoglobin [23,26,37].

One of the principal obstacles for this identification is that functional redundancy among factors regulating the cyt *c* content has limited the effectiveness of genetic analysis. To circumvent this, we managed to eliminate the influence of the cAMP-CRP regulatory system on the cyt *c* content, which functions as the master regulator in the respiration of *S. oneidensis* [35,45,46]. As nitrite compromises cAMP biosynthesis by a yet-unknown mechanism, it intertwines with the regulatory system in terms of the cyt *c* content [26,34]. By using a cAMP-independent Crp variant, we substantiated that nitrite/NO could cause a reduction in the cyt *c* content that is independent of cAMP-Crp. cyt *c* is formed by establishing a covalent linkage between two precursors, apocyt *c* and heme [2]. Given that cAMP-Crp impacts the cyt *c* content by directly mediating the transcription of a large number of cyt *c* genes, heme becomes a potential target whose homeostasis may be influenced by nitrite/NO. Our results support the notion that intracellular heme levels are not responsive to the addition of either nitrite or NO. This is not surprising as it has been well established that NO acts to block cellular heme insertion into a broad range of hemoproteins by either direct or indirect means, without affecting heme availability within the cells [60,61]. Nevertheless, heme at elevated concentrations was found to be able to relieve, at least partially, the suppression of nitrite/NO on the cyt *c* content. This antagonistic effect of heme on nitrite/NO inhibition has led to the identification of CcmE as a common target of nitrite and NO.

Although cyt *c* biosynthesis takes place in the periplasm, both precursors, apocyt *c* and heme, are generated in the cytoplasm [2]. Compared to the well-understood apocyt transmembrane transportation, heme delivery presents an interesting challenge for bacteria, which have to carry out multiple steps promptly, including intracellular heme traffic from the synthesized site to the CcmABC complex, heme transport through the inner membrane by CcmC, and heme transfer from CcmC to CcmE and, eventually, to CcmF [11,62,63,64]. Our data presented here suggest that the steps in which CcmE is involved are likely under the influence of nitrite/NO. When NO, presumably nitrite too, binds to heme, it typically decreases the strength of the axial bond formed between the heme iron and a coordinating protein residue, weakening or even disrupting the bond [17,65]. Clearly, our data demonstrated that the covalent bonds within cyt *c* and CcmE, unlike those non-covalent ones within other hemoproteins, may not be disrupted by nitrite/NO at physiological relevant concentrations. However, the impacts of nitrite/NO on CcmE are apparent. The shifts in the UV–visible spectra of CcmE upon the treatment indicate that the protein is modified by both agents. The consequence of the modification is highly likely to form nitro coordination between nitrite and the protein, as revealed by a recent work that employed a variant of *alcaligenes xylosoxidans* cyt *c*, which displays a remarkably increased heme affinity for nitrite [41].

In addition, we observed that the amount of CcmE detectable by heme staining reduces significantly. We do not yet know the underlying mechanism behind this. It is possible that the Fe(III)-nitrite complex is resistant to heme staining agents because of the loss of the redox activity. Meanwhile, it would be imprudent to exclude the possibility that these nitrogen oxides interfere with heme transfer from CcmC to CcmE. If this occurs, a portion of CcmE would exist in heme-free form, which has been shown to be inserted into the membrane and stable comparable to the heme-bound counterpart [13,66,67]. We are working to test this possibility.

The data presented also reveal an evident difference in the sensitivity to nitrite/NO between cyt *c* and CcmE. CcmE is substantially more susceptible to modification by nitrite/NO than cyt *c*, at least by four-fold. The enhanced susceptibility of CcmE to nitrite/NO explains why CcmE is inhibited by nitrite/NO in the presence of a large repertoire of cyts *c* in *S. oneidensis*. We envision that this modification impairs CcmE activity, thus lowering its heme delivery efficiency. While further investigations into this are underway, our current findings have highlighted a new mechanism by which nitrite/NO interfere with CcmE-mediated heme transfer to compromise cyt *c* maturation.

## 4. Materials and Methods

### 4.1. Bacterial Strains, Plasmids, and Culture Conditions

All bacterial strains and plasmids used in this study are listed in Table 1, and information about all of the primers used in this study is available upon request. All chemicals were obtained from Sigma (Shanghai, China), unless specifically noted. For genetic manipulation, *E. coli* and *S. oneidensis* were grown aerobically in lysogeny broth (LB, Difco, Detroit, MI, USA) at 37 and 30 °C. When appropriate, chemicals at the following concentrations were added to the growth medium: 2,6-diaminopimelic acid (DAP), 0.3 mM; kanamycin (Kan), 50 μg/mL; gentamycin (Gent), 15 μg/mL; catalase on plates, 2000 U/mL.

Growth of *S. oneidensis* strains under aerobic or anaerobic conditions was measured by recording the optical density of cultures at 600 nm (OD_600_). Defined medium MS containing 30 mM lactate as the electron donor and 20 mM TMAO or 20 mM fumarate as electron acceptor were used as previously described [68]. For aerobic growth, mid-log phase cultures were inoculated into fresh medium to an OD_600_ of ~0.05 and shaken at 200 rpm at 30 °C. For anaerobic growth, mid-log phase aerobic cultures were pelleted by centrifugation, purged with nitrogen, and suspended in fresh media prepared anaerobically to an OD_600_ of ~0.05.

### 4.2. Construction of in-Frame Deletion Strains

In-frame deletion strains for *S. oneidensis* were constructed according to the *att*-based Fusion PCR method, as described previously [5]. In brief, two fragments flanking the gene of interest were amplified by PCR, which were linked by the second round of PCR. The fusion fragments were integrated into plasmid pHGM01 by using Gateway BP clonase II enzyme mix (Invitrogen) according to the manufacturer’s instruction. The resultant plasmid was transformed by electroporation into *E. coli* WM3064, and the verified ones were transferred to *S. oneidensis* strains by conjugation. Integration of the mutagenesis constructs into the chromosome was selected by resistance to gentamycin and confirmed by PCR. Verified trans-conjugants were grown in LB in the absence of NaCl and plated on LB supplemented with 10% sucrose. Gentamycin-sensitive and sucrose-resistant colonies were screened by PCR for deletion of the target gene. Mutants were verified by sequencing the site for the intended mutation.

### 4.3. Site-Directed Mutagenesis

Site-directed mutagenesis was performed to generate Crp proteins carrying point mutations using a QuikChange II XL site-directed mutagenesis kit (Stratagene) [69]. The *crp* gene within pHGEN-P*tac* was subjected to modification, and the resulting products were digested by *Dpn*I at 37 °C for 6 h and subsequently transformed into *E. coli* WM3064. The vectors carrying the intended mutations, which was verified by sequencing, were transferred into the relevant *S. oneidensis* strains by conjugation.

### 4.4. Heme c Assays

Cultures of *S. oneidensis* strains grown in liquid medium to the early stationary phase were centrifuged, and the pellets were photographed. The cyt *c* abundance of strains was first estimated by the color intensity of the cell pellets. Subsequently, the pellets were suspended in PBS, adjusted to the same OD_600_ values, and the cells from the same-volume aliquots were disrupted. All proteins were precipitated by trichloroacetic acid precipitation [70] and assayed for heme *c* levels with the QuantiChrom heme assay kit (BioAssay Systems) according to the manufacturer’s instructions. If necessary, the values were normalized to the concentrations of the total proteins for each sample, which were determined by the bicinchoninic acid assay (Pierce Chemical) throughout this study.

### 4.5. Controlled Gene Expression and Complementation of Mutants

Controlled gene expression was used in genetic complementation of mutants and assessment of the physiological effects of proteins at varying levels. The genes of interest were generated by PCR, cloned into plasmid pHGEN-Ptac under the control of Isopropyl β-D-1-thiogalactoside (IPTG)-inducible promoter Ptac, and the resultant vectors were transformed into *E. coli* WM3064 [71]. After verification by sequencing, the vectors were transferred into the relevant *S. oneidensis* strains via conjugation. Expression of the cloned genes was controlled by IPTG at varying concentrations.

### 4.6. Preparation of Ether-Treated Bacterial Cells

*S. oneidensis* cells grown in LB under aerobic conditions to the early stationary phase (~0.8 of OD_600_) were harvested by centrifugation. From this sediment, ether-treated bacteria were prepared essentially as described by Mirelman et al. [72]. In brief, suspended cell samples of 5 mL (~0.8 of OD_600_) were mixed with ether under the conditions specified by Vosberg and Hoffmann-Berling [42]. After removal of the ether layer, the cells in the aqueous medium were sedimented (7000× *g*, 8 min), and the pellet was resuspended in basic medium at a concentration of approximately 10 mg of protein/mL (1010 cells/mL) and stored at −20 °C.

### 4.7. Expression and Purification of Recombinant CcmE

The DNA fragment containing the sequences for the *ccmE* gene and for His_6_-tag was cloned into pHGEN-Ptac. *S. oneidensis* Δ*ccmE* carrying the resulting vector grown in LB to the mid-log phase was induced with 1 mM IPTG at 16 °C for 8 h and harvested by centrifugation. Cells were resuspended in Tris-HCl buffer (10 mM Tris-HCl, 150 mM NaCl, 1 mM EDTA, pH8.0), supplemented with 1 mM PMSF and 40 μg/mL DNase I (Roche). Cells were lysed by a high-pressure cell disruptor (JNBIO), cleared of cell debris by centrifugation at 20,000× *g* for 30 min at 4 °C, and followed by the separation of soluble and membrane fractions via high-speed ultracentrifugation at 160,000× *g* for 45 min at 4 °C. Membrane pellets were solubilized in Tris-HCl buffer with 1% n-dodecyl-β-D-maltopyranoside (Anatrace) and affinity purified by Beads IDA-Nickel (Solarbio). The subsequent purification steps were performed according to the manufacturer’s instructions. The purified proteins were analyzed by 12% SDS-PAGE, followed by staining with Coomassie Brilliant Blue R250.

### 4.8. SDS-PAGE, Heme-Staining, and Western Blotting

Mid-log phase cells were harvested, washed with phosphate-buffered saline (PBS), resuspended in the same buffer, and sonicated. cyt *c* from bovine heart (Mol. Wt. 12,327) was acquired from Sigma-Aldrich Co. Protein concentrations of the cell lysates were determined by the bicinchoninic acid assay (Pierce Chemical). For heme staining, the cell lysates were separated by SDS-PAGE using 12% polyacrylamide gels and stained with 3,3′,5,5′-tetramethylbenzidine (TMBZ), as described elsewhere [73]. His_6_-tagged recombinant CcmE was examined by Western blotting as described before [74]. Cell extracts for the defection of CcmE were subjected to electrophoresis on 10% SDS polyacrylamide gels (PAGE). Proteins were transferred to polyvinylidene difluoride (PVDF) membranes for 1 h at 60 V using a Criterion blotter (Bio-Rad). The blotting membrane was probed with antibodies against the His_6_ tag (Sangon Biotech, Shanghai, China), followed by a 1:10,000 dilution of goat anti-rabbit immunoglobulin G-alkaline phosphatase conjugate. The alkaline phosphatase was detected using a chemiluminescence Western blotting kit (Roche Diagnostics) in accordance with the manufacturer’s instructions. Images were visualized with Clinx Imaging System (Clinx, Shanghai, China).

### 4.9. Spectroscopic Analysis

To explore the effects of nitrite and NO on cyts *c* and purified CcmE, UV–visible spectra were recorded on a Varioskan Flash (Thermo scientific, Shanghai, China). Protein was incubated with various concentrations of nitrite and NOC18 at room temperature for 20 min before the measurements were taken.

### 4.10. β-Galactosidase Activity Assay

The activity of the promoters of interest was assessed using a single-copy integrative *lacZ* reporter system as described previously [75]. The sequence in sufficient length (~400 bp) upstream of the gene of interest was amplified and inserted in front of the full-length *E. coli lacZ* gene in plasmid pHGEI01. The resulting plasmid was verified by sequencing, introduced into *E. coli* WM3064, and then, conjugated with relevant *S. oneidensis* strains. Once transferred into *S. oneidensis* strains, pHGEI01 containing the promoter of interest integrated into the chromosome, and the antibiotic marker was then removed by an established approach [32]. Cultures of the mid-exponential phase were collected by centrifugation, washed with PBS, and treated with lysis buffer (0.25 M Tris/HCl, 0.5% Triton X-100, pH 7.5). Extracts were collected by centrifugation and applied for the enzyme assay by adding o-nitrophenyl-β-D-galactopyranoside (ONPG) (4 mg/mL). Changes in absorption over time were monitored at 420 nm with a Synergy 2 Pro200 Multi-Detection Microplate Reader (Tecan), and the results are presented as Miller units.

### 4.11. Other Analyses

To estimate the relative abundance of the proteins in the gels, the intensities of the bands were quantified using ImageJ software [76]. Student’s *t-*test was performed for pairwise comparisons. In the figures, the values are presented as the means +/− the standard deviation (SD).

**Table 1 ijms-23-13841-t001:** Strains and plasmids used in this study.

Strain or Plasmid	Description	Source/Reference
*E. coli*		
DH5α	Host strain for plasmids	Laboratory stock
WM3064	Donor strain for conjugation; Δ*dapA*	W. Metcalf, UIUC ^a^
*S. oneidensis*		
MR-1	Wild type	ATCC 700550
HG0624	Δ*crp* derived from MR-1	[45]
HG0696	Δ*ccmF* derived from MR-1	[5]
HG0958	Δ*nrfA* derived from MR-1	[58]
HG1070	Δ*ccmE* derived from MR-1	This study
HG3901-0624	Δ*cya*Δ*crp* derived from MR-1	This study
Plasmids		
pHGM01	Ap^r^ Gm^r^ Cm^r^ suicide vector	[5]
pHGEI01	Integrative *lacZ* reporter vector	[75]
pHGEN-P*tac*	IPTG-inducible P*tac* expression vector	[71]
pHGEI01-P*_hemA_*	For measuring P*_hemA_* activity	[46]
pHGEI01-P*_hemC_*	For measuring P*_hemC_*activity	[46]
pHGEI01-P*_hemH1_*	For measuring P*_hemH1_* activity	[77]
pHGEI01-P*_hemH2_*	For measuring P*_hemH2_* activity	[77]
pHGEI01-P_CC(−41.5)_	For measuring P_CC(−41.5)_ activity	This study
pHGEI01-P*_cyd_*	For measuring P*_cyd_* activity	This study
pHGEN-P*tac*-*Socrp*	For inducible production of *S. oneidensis* Crp	This study
pHGEN-P*tac*-*Eccrp*	For inducible production of *E. coli* Crp	This study
pHGEN-P*tac*-*Eccrp**	For inducible production of cAMP-independent *E. coli* Crp	This study
pHGEN-P*tac*-*hemA*	For inducible production of HemA	This study
pHGEN-P*tac*-*ccmI*	For inducible production of CcmI	This study
pHGEN-P*tac*-*ccmABCDE*	For inducible production of CcmABCDE	This study
pHGEN-P*tac*-*ccmABCD*	For inducible production of CcmABCD	This study
pHGEN-P*tac*-*ccmFGH ccmABCD*	For inducible production of CcmFGH	This study
pHGEN-P*tac*-*ccmE*	For inducible production of *S. oneidensis* CcmE	This study
pHGEN-P*tac*-*EcccmE*	For inducible production of *E. coli* CcmE	This study
pHGEN-P*tac*-*PaccmE*	For inducible production of *P. aeruginosa* CcmE	This study
pHGEN-P*tac*-*ccmE^his^*	For inducible expression of His-tagged CcmE	This study

^a^ UIUC, University of Illinois Urbana-Champaign.

## Figures and Tables

**Figure 1 ijms-23-13841-f001:**
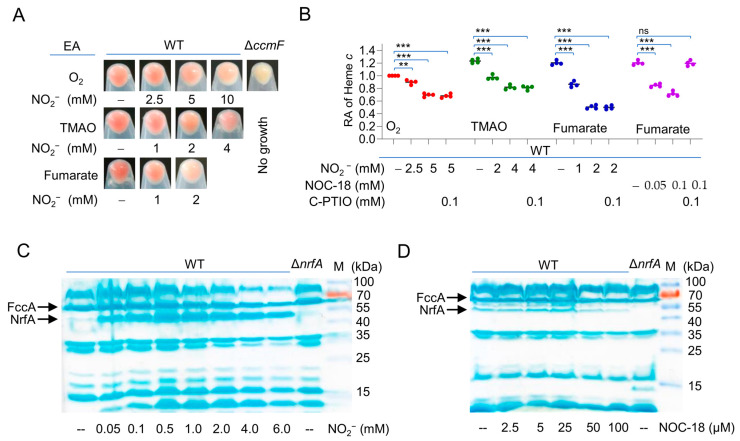
Nitrite and NO down-regulate cyt *c* content of *S. oneidensis*. (**A**) The cell color phenotypes. Shown are cell pellets of indicated strains grown on one of the EAs, O_2_, TMAO, and fumarate, to the early stationary phase. Strains include the wild-type (WT) and cyt-*c*-deficient ∆*CcmF*. (**B**) Heme *c* levels. Samples prepared as in (**A**) were lysed for quantification of heme *c*. NOC-18 and C-PTIO were used as the NO producer and scavenger, respectively. The data were first adjusted to the protein levels of the samples, and then, the averaged heme *c* levels of the mutants were normalized to that in the WT, which was set to 1, giving the relative abundance (RA). Asterisks indicate statistically significant difference of the values compared (*n* = 4; ns, not significant; **, *p* < 0.01; ***, *p* < 0.001). (**C**,**D**) Heme staining. Samples prepared the same as for the quantification of heme *c* were processed; the protein contents were quantified, and equal amounts of the proteins were separated in SDS-PAGE, then subjected to heme staining. M, molecular weight marker. FccA and NrfA are indicated by arrows. All experiments were performed three times, and representatives as in (**A**,**C**,**D**) are presented.

**Figure 2 ijms-23-13841-f002:**
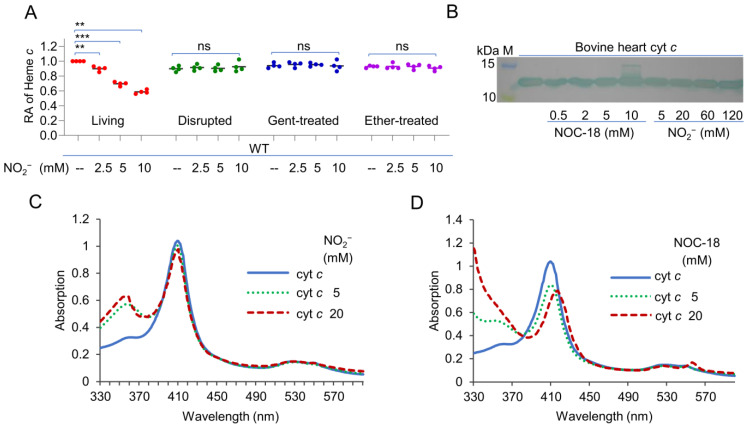
Nitrite and NO compromise the cyt *c* content in growing cells only. (**A**) Heme *c* levels. Cells at the early stationary phase were collected and processed by sonication or killed without destroying the cell morphology by treated with gentamycin or ether. All these samples were incubated with nitrite at varying concentrations for 5 h. Quantification of heme *c* was performed, and the data were processed as described in Figure 1b. Asterisks indicate statistically significant difference of the values compared (*n* = 4; ns, not significant; **, *p* < 0.01; ***, *p* < 0.001). (**B**) Heme staining. Bovine heart cyt *c* (2 μM) was incubated with nitrite and NO at the indicated concentrations for 5 h and then was examined by SDS-PAGE/heme staining. M, molecular weight marker. (**C**,**D**) UV–visible spectra of 5 μM cyt *c* solutions. Shown are untreated and treated with nitrite **C** or NOC-18 (**D**) at indicated concentrations for 5 h. In (**B**–**D**), experiments were performed three times, and a representative as in (**B**) and the averaged as in (**C**,**D**) are presented.

**Figure 3 ijms-23-13841-f003:**
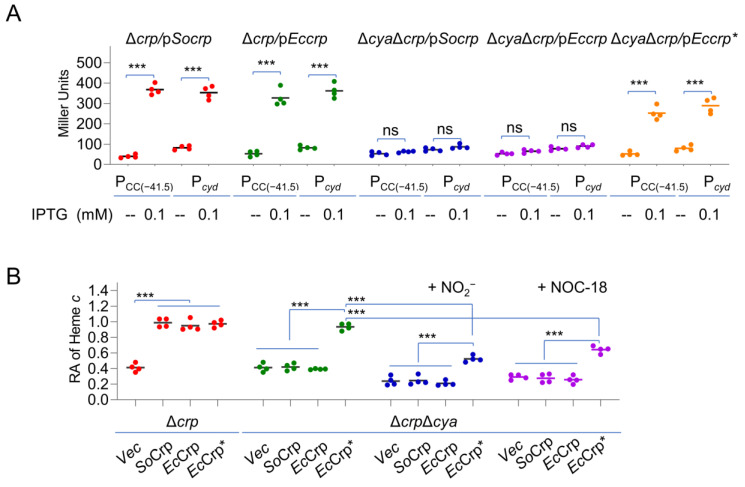
cAMP-CRP is not exclusively responsible for the nitrite-/NO-mediated reduction in the cyt *c* content. (**A**) *Ec*Crp^T128L-S129I^ (encoded by *Eccrp**) is independent of cAMP in *S. oneidensis*. Production of *So*Crp, *Ec*Crp, and *Ec*Crp^T128L-S129I^ was driven by the IPTG-inducible promoter in the *S. oneidensis crp* and *crp*-*cya* (missing all enzymes for cAMP synthesis) mutants. P_CC(−41.5)_ and P*_cyd_* are promoters that are directly controlled by *Ec*Crp and *So*Crp, respectively. Promoter activity in cells grown to the early stationary phase was determined by LacZ reporters. (**B**) Heme *c* levels in the *S. oneidensis crp* and *crp*-*cya* mutants producing *So*Crp, *Ec*Crp, and *Ec*Crp^T128L-S129I^ with IPTG at 0.1 mM. Effects of 5 mM nitrite and 0.1 mM NOC-18 on *Ec*Crp^T128L-S129I^ were compared. Vec, empty vector. Asterisks indicate statistically significant difference of the values compared (*n* = 4; ns, not significant; ***, *p* < 0.001).

**Figure 4 ijms-23-13841-f004:**
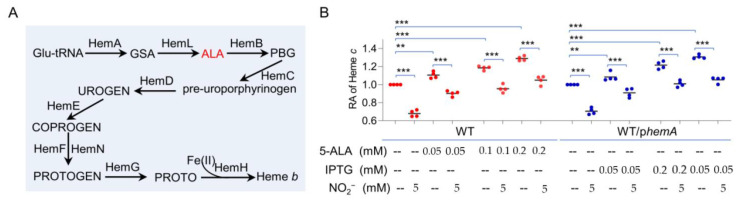
Heme antagonizes inhibitory effects of nitrite on the cyt *c* content. (**A**). Biosynthesis pathway of heme *b* in *S. oneidensis*. GSA, glutamate-1-semialdehyde; ALA, 5-aminolevulinic acid; PBG, porphobilinogen; UROGEN, uroporphyrinogen III; CORROGEN, coproporphyrinogen III; PROTOGEN, protoporphyrinogen IX; PROTO, protoporphyrin IX. (**B**). Heme *c* levels were quantified and are presented as relative abundance (RA). 5-ALA, the intermediate dictating the heme biosynthesis rate. Production of HemA was driven by the IPTG-inducible promoter in *S. oneidensis*. Asterisks indicate statistically significant difference of the values compared (*n* = 4; ns, not significant; **, *p* < 0.01; ***, *p* < 0.001).

**Figure 5 ijms-23-13841-f005:**
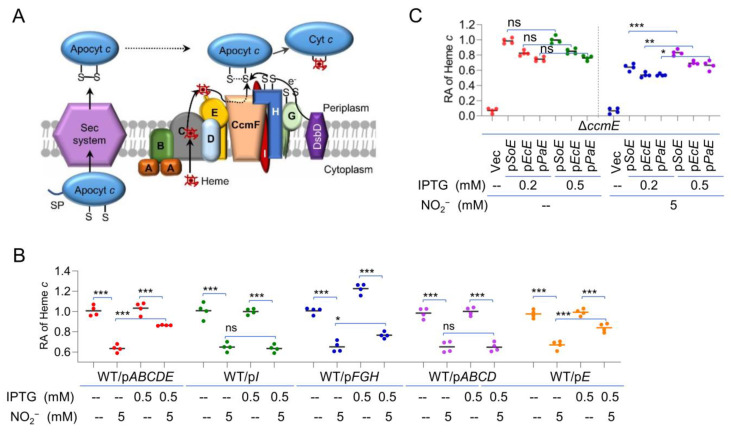
CcmE is a likely target of nitrite/NO in cyt *c* production. (**A**) A simplified model for cyt *c* biosynthesis in *S. oneidensis*. Both apocyt *c* and intracellular heme are transported to the periplasm, where heme attachment occurs. CcmABC transports heme across the membrane and delivers it to CcmE. CcmE then delivers heme to CcmF, which catalyzes heme attachment. (**B**) Heme *c* levels in WT overproducing Ccm components driven by the IPTG-inducible promoter without or with 5 mM nitrite. (**C**) Heme *c* levels in the *ccmE* mutant overproducing CcmE of *S. oneidensis* (*SoE*), *E. coli* (*EcE*), and *P. aeruginosa* (*PaE*) driven by the IPTG-inducible promoter without or with 5 mM nitrite. Asterisks indicate statistically significant difference of the values compared (*n* = 4; ns, not significant; *, *p* < 0.05; **, *p* < 0.01; ***, *p* < 0.001).

**Figure 6 ijms-23-13841-f006:**
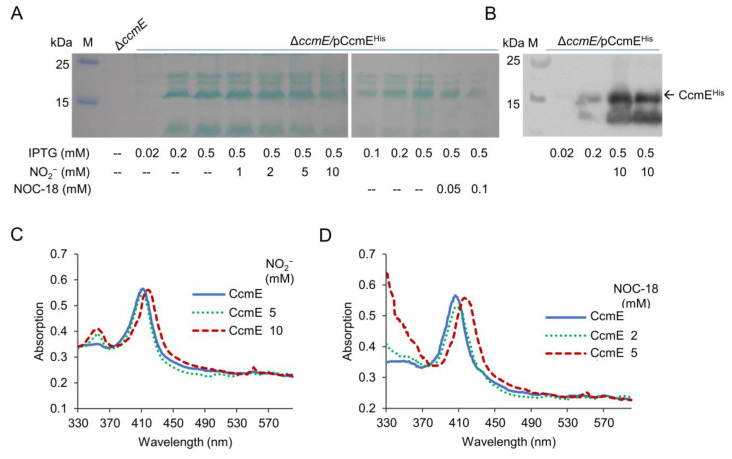
Impacts of nitrite/NO on CcmE. (**A**,**B**) Impacts of nitrite/NO on CcmE in vivo. Δ*ccmE* cells producing His-tagged CcmE with IPTG at varying concentrations were grown to the early stationary phase without or with indicated nitrogen oxides. Proteins with heme covalently attached were examined by heme staining (**A**) and by Western blotting (**B**). (**C**,**D**) UV–visible spectra of 2 μM CcmE solutions. Shown are untreated and treated with nitrite (**C**) or NOC-18 (**D**) at indicated concentrations for 5 h. Experiments were performed three times, and representatives are presented.

## Data Availability

Not applicable.

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
