# Peer review of "A Common Target of Nitrite and Nitric Oxide for Respiration Inhibition in Bacteria"

_ijms, 2022, doi:10.3390/ijms232213841_

Round 1
Reviewer 1 Report
In the paper entitled "A common target of nitrite and nitric oxide for respiration inhibition in bacteria", the authors prove, using Shewanella oneidensis as model, a common mechanism by which nitrite and NO inhibit bacteria respiration, by compromising its cyt c biosynthesis.
The research done is complex, the paper is well organised and written. Therefore, I believe that it can be accepted for publication in IJMS, after some very minor changes:
- all abbreviations should be defined when first mentioned within the text;
- page 10 - line 39 (there is a problem with row numbering, so be careful): replace "The date presented also reveal" with "The data presented also reveal";
- reference no. 55: the year of publication should be in Bold face;
- regarding the bibliography, almost 50% of the references cited are more than 10 years old...I believe that a more up to date bibliography is advisable.
Author Response
In the paper entitled "A common target of nitrite and nitric oxide for respiration inhibition in bacteria", the authors prove, using Shewanella oneidensis as model, a common mechanism by which nitrite and NO inhibit bacteria respiration, by compromising its cyt c biosynthesis.
The research done is complex, the paper is well organised and written. Therefore, I believe that it can be accepted for publication in IJMS, after some very minor changes:
- all abbreviations should be defined when first mentioned within the text;
Response: We defined all abbreviations when first mentioned.
- page 10 - line 39 (there is a problem with row numbering, so be careful): replace "The date presented also reveal" with "The data presented also reveal";
Response: We corrected it.
- reference no. 55: the year of publication should be in Bold face;
Response: We corrected it.
- regarding the bibliography, almost 50% of the references cited are more than 10 years old...I believe that a more up to date bibliography is advisable.
Response: We have tried our best to cite the most recently published articles. But for this field, a large number of classical studies that we have to cite were done a while ago.
Reviewer 2 Report
Dear authors,
In this paper the authors investigate the existence of a common mechanism by which nitrite and NO influence bacterial physiology. Using S. oneidensis as model organism, an organism with a large amount of cyt c. Arriving as the main conclusion that both nitrite and NO interferes with heme content through altering CcmE, which causes a decrease in the synthesis of cyt c.
I have enjoyed reading this work very much. This paper has a great biochemical research value, the authors have made a great effort and show a large amount of data that notably support their hypotheses. They have been able to solve the great difficulties and obstacles encountered with great ingenuity. Such as the use of cAMP-independent Crp variant. I only have a few minor considerations to say. The introduction is very well written, makes a good state of the art of this topic, and deals with the matter in due depth. The materials and methods are very well written and in sufficient detail.
Minors:
I miss that the introduction does not mention the mechanisms that bacteria have for nitrite and NO detoxification; I suggest the authors make reference to the main ones.
Results
L88: “or an NO-producing Shewanella strain to levels below measurable limit” I'm sorry I don't understand this sentence, please rewrite it
L91: “eliminating the possibility that the observed effect of nitrite on the cyt c content is via NO” I do not think that this statement can be taken as correct without a doubt, when growing the cells with nitrite, they are long experiments, and the production of NO can be higher than that which the cPTIO can block, please consider this.
Fig 1B: In the foot of the figure, the amount of NOC-18 mM used is not indicated.
Fig 1C. What is the identity of the other bands that appear on the gel, have you identified them? because apart from FccA and NtfA it seems to me that there are other bands that are also affected, and their identification may be interesting.
Fig 2. Please indicate in the figure that the Gent-treated and the Ether-treated are non-viable cells. Please indicate the concentration of Bovine heart cyt c utilized.
Fig 3. Please, for greater clarity, indicate what it means (Vec)
L224: “that that”
Fig 5. Please, in the figure legend indicate what it means pSoE, pEcE, ……
Discussion:
I suggest the authors comment on the mechanisms bacteria have to reduce nitrite to NO, and whether such enzymes are present in S. oneidensis.
L447 “All proteins were precipitated by trichloroacetic acid precipitation” Please specify how or indicate the reference
L495 “His6 tag antibodies” please, Indicate the manufacturer
L542: “☆” please remove
Author Response
In this paper the authors investigate the existence of a common mechanism by which nitrite and NO influence bacterial physiology. Using S. oneidensis as model organism, an organism with a large amount of cyt c. Arriving as the main conclusion that both nitrite and NO interferes with heme content through altering CcmE, which causes a decrease in the synthesis of cyt c.
I have enjoyed reading this work very much. This paper has a great biochemical research value, the authors have made a great effort and show a large amount of data that notably support their hypotheses. They have been able to solve the great difficulties and obstacles encountered with great ingenuity. Such as the use of cAMP-independent Crp variant. I only have a few minor considerations to say. The introduction is very well written, makes a good state of the art of this topic, and deals with the matter in due depth. The materials and methods are very well written and in sufficient detail.
Minors:
I miss that the introduction does not mention the mechanisms that bacteria have for nitrite and NO detoxification; I suggest the authors make reference to the main ones.
Response: Thanks for this suggestion. We introduced this point in the introduction briefly. .
Results
L88: “or an NO-producing Shewanella strain to levels below measurable limit” I'm sorry I don't understand this sentence, please rewrite it
Response: We rephrased the sentence.
L91: “eliminating the possibility that the observed effect of nitrite on the cyt c content is via NO” I do not think that this statement can be taken as correct without a doubt, when growing the cells with nitrite, they are long experiments, and the production of NO can be higher than that which the cPTIO can block, please consider this.
Response: Thanks for this meticulous comment. According to many studies and our experience, 0.1 mM c-PTIO can completely block the NO signal produced by NOC-18.
Fig 1B: In the foot of the figure, the amount of NOC-18 mM used is not indicated.
Response: We corrected it.
Fig 1C. What is the identity of the other bands that appear on the gel, have you identified them? because apart from FccA and NtfA it seems to me that there are other bands that are also affected, and their identification may be interesting.
Response: Thanks for the comment. Shewanella oneidensis MR-1 contains 42 cyt c. nitrite and NO above certain concentrations showed a negative effect on the abundance of all cyts c. As for the bands other than FccA and NrfA, including a variety of terminal reductase and electron transfer protein.
Fig 2. Please indicate in the figure that the Gent-treated and the Ether-treated are non-viable cells. Please indicate the concentration of Bovine heart cyt c utilized.
Response: We revised the legend to include these.
Fig 3. Please, for greater clarity, indicate what it means (Vec)
Response: We indicated it.
L224: “that that”
Response: We corrected it.
Fig 5. Please, in the figure legend indicate what it means pSoE, pEcE, ……
Response: We indicated them.
Discussion:
I suggest the authors comment on the mechanisms bacteria have to reduce nitrite to NO, and whether such enzymes are present in S. oneidensis.
Response: Thanks for this comment. We concisely discussed this in the discussion.
L447 “All proteins were precipitated by trichloroacetic acid precipitation” Please specify how or indicate the reference
Response: We added a reference about trichloroacetic acid precipitation.
L495 “His6 tag antibodies” please, Indicate the manufacturer
Response: We indicated it.
L542: “☆” please remove
Response: We removed it.
Reviewer 3 Report
The first question is whether peroxynitrite is formed in the incubation medium and in Shewanella oneidensis when adding nitrite and NOC-18? The second question is whether tyrosine derivatives are formed in the proteins of Shewanella oneidensis? The third question is, do the authors have data on the level of nitrosothiols RSNO in Shewanella oneidensis cells?
Author Response
The first question is whether peroxynitrite is formed in the incubation medium and in Shewanella oneidensis when adding nitrite and NOC-18? The second question is whether tyrosine derivatives are formed in the proteins of Shewanella oneidensis? The third question is, do the authors have data on the level of nitrosothiols RSNO in Shewanella oneidensis cells?
Response: Thanks for these comments. Our physiological and biochemical work presented here does not focus on how nitrite and NO are transformed in the cell and whether and what reactive nitrogen species are formed. Even we have worked on this subject for 15 years, we never try to figure out the intermediates formed from the addition of nitrite and NOC-18. So we do not know the answers to these questions. But our knowledge built on these studies has suggested that the chance is small. We have previously identified HCO to be the primary target of nitrite and cytochrome c proteins to be a sink for NO. As these proteins are hemoproteins, it would be reasonable to assume that they are more likely to interact with nitrogen oxides to form Fe-NO adducts.